# Mesenchymal Stem Cell-Derived Exosomes and Intervertebral Disc Regeneration: Review

**DOI:** 10.3390/ijms23137306

**Published:** 2022-06-30

**Authors:** Basanta Bhujel, Hae-Eun Shin, Dong-Jun Choi, Inbo Han

**Affiliations:** 1Department of Biomedical Science, College of Life Sciences, CHA University, Seongnam-si 13496, Korea; basantabhujel86@gmail.com (B.B.); tlsgodms223@naver.com (H.-E.S.); 2Department of Medicine, CHA Univerity School of Medicine, Seongnam-si 13496, Korea; kenchoi94@naver.com; 3Department of Neurosurgery, CHA University School of Medicine, CHA Bundang Medical Center, Seongnam-si 13496, Korea

**Keywords:** lower back pain, mesenchymal stem cell, exosomes, intervertebral disc degeneration, regeneration

## Abstract

Intervertebral disc degeneration (IVDD) is a common cause of lower back pain (LBP), which burdens individuals and society as a whole. IVDD occurs as a result of aging, mechanical trauma, lifestyle factors, and certain genetic abnormalities, leads to loss of nucleus pulposus, alteration in the composition of the extracellular matrix, excessive oxidative stress, and inflammation in the intervertebral disc. Pharmacological and surgical interventions are considered a boon for the treatment of IVDD, but the effectiveness of those strategies is limited. Mesenchymal stem cells (MSCs) have recently emerged as a possible promising regenerative therapy for IVDD due to their paracrine effect, restoration of the degenerated cells, and capacity for differentiation into disc cells. Recent investigations have shown that the pleiotropic effect of MSCs is not related to differentiation capacity but is mediated by the secretion of soluble paracrine factors. Early studies have demonstrated that MSC-derived exosomes have therapeutic potential for treating IVDD by promoting cell proliferation, tissue regeneration, modulation of the inflammatory response, and reduced apoptosis. This paper highlights the current state of MSC-derived exosomes in the field of treatment of IVDD with further possible future developments, applications, and challenges.

## 1. Introduction

Lower back pain (LBP) is a major symptom derived from intervertebral disc degeneration (IVDD), and LBP places a significant burden not only on individuals but on society as well [1,2]. The exact cause of IVDD is not fully known; however, it is influenced by genetic factors, mechanical loading, infection, nutritional factors, and aging. Thus, it is considered a complex and multifactorial process [3]. At present, physical therapy, pharmacological interventions and surgery are the main clinical treatments for LBP, but these approaches mainly target temporary symptomatic relief rather than targeting the pathogenesis. These modalities are not fully capable of restoring the structural and functional properties of the degenerated disc [4]. Various new approaches such as the use of growth factors, biologics, gene transfection, and mesenchymal stem cells (MSCs) are being used in both preclinical and clinical studies to treat degenerated intervertebral discs (IVD) [5]. Among them, cell-based therapy, both allogeneic and autologous, is receiving increased interest due to its ability to target numerous pathways that induce IVDD. However, challenges persist in achieving a high survival rate of transplanted cells; therefore, this technique is burdensome for researchers in terms of controlling cell viability and differentiation [3,6,7].

In the present context, MSCs are considered the most appealing and ideal cell source for IVD regeneration both in vitro and in vivo. The potential of MSCs for treating degenerated discs has been indicated in many preclinical studies and, more recently, in clinical trials [8,9,10,11]. Studies have demonstrated the ability of MSCs to differentiate into a nucleus pulposus (NP)-like phenotype to enhance the matrix production of glycosaminoglycans (GAGs), increase disc height, and improve hydration in the IVD [12]. Similarly, studies have shown that when MSCs are co-cultured with NP cells (NPCs), they communicate in a bidirectional manner with NPCs; therefore, in addition to undergoing differentiation and de novo synthesis of the extracellular matrix (ECM), implanted cells may influence NPC function through the secretion of bioactive factors such as anabolic growth factors [13,14]. Further, MSCs also possess anti-inflammatory and anti-catabolic properties, which could reduce cytokine levels, thereby modulating the inflammatory niche to generate a healthier, non-degenerative phenotype in native NPCs [12].

Studies on the mechanism of MSC-based therapies have offered increasing proof that exosomes are crucial biological structures that help in intercellular communication, and take part in various physiological and pathological functions [15]. They exchange various biomolecules such as proteins and nucleic acids from the parent cells to the receiver cells and participate in the body’s immune response, cell migration, cell differentiation, and so on [16,17]. In particular, stem cell-derived exosomes represent an important opportunity for the safe and effective treatment of IVDD due to their ability to retain the therapeutic benefits of their origin cells without the risk associated with stem cell-based therapies. At this present time, the research on exosomes derived from MSCs in many clinical conditions is gaining popularity. Their role in regenerative medicine is increasing, as they can tolerate a harsh environment and assist in the regeneration of NPCs, inhibit apoptosis, modulate the immune response, stimulate angiogenesis, downregulate inflammatory cytokines, restore ECM, and help in the maintenance of homeostasis of the IVD [18,19,20,21,22]. For this review, we searched for the keywords “intervertebral disc”, “exosomes,” and “mesenchymal stem cells” in PubMed (https://pubmed.ncbi.nlm.nih.gov/ (accessed on 1 May 2022) and Google Scholar from February 2007 to April 2022.

## 2. Pathophysiology of IVDD

IVDs are embedded between the vertebrae of the spine and provide flexibility and elasticity to the spine. The structure of the intervertebral disc consists of the NP, which is encapsulated by annulus fibrosis (AF), and two cartilaginous endplates [23]. The major cause of IVDD is disruption of homeostasis within the disc, resulting in a pathological condition [24]. The crucial factor for the breakdown of the IVD is loss of the disc cells and alternation in the ECM, which may cause weakening of tissues and changes in the morphology and functions of cells [4,25,26,27]. Furthermore, nutritional, environmental, and genetic factors are interlinked with IVDD [28,29]. These factors are associated with changes in cellular morphology, the induction of inflammation, a rapid increment of senescent cells, apoptosis, the secretion of pro-inflammatory cytokines, and autophagy in the IVD [30,31,32,33,34]. The presence of high catabolism and low anabolism in the ECM is usually associated with IVDD. The release of the inflammatory cytokines such as tumor necrosis factor-α (TNF-α) and interleukin-1β are the major inflammatory response that takes place during IVDD. Similarly, senescent cells produce senescence-associated secretory phenotypes. As a result, an increment in the secretion of pro-inflammatory cytokines, chemokines, and tissue-damaging protease takes place, which causes degeneration of the disc [35,36]. Likewise, a change in the nutritional level in the disc may cause a reduction of oxygen supply and a decrease in the pH in the disc, which causes degradation in the ECM of the IVD [32]. Some of the important environmental factors that contribute to IVDD are smoking, trauma, excessive calories leading to obesity, and mechanical loading. Similarly, aging may disrupt the disc ECM; as a result, a reduction in aggrecan takes place in the disc NP with a decrease in the disc hydration. As dehydrated NPs are unable to balance the forces in the adjacent vertebrae, this may lead to the destruction of the structural and functional mechanism in the disc. This results in the annular tear and disc herniation in the NP [31]. Furthermore, in IVDD, inherited components also play a significant role and are associated with the genetic changes in the matrix proteins. Polymorphisms in the genes encoding aggrecan, collagen I and XI, and the interleukin-1 signaling proteins have been found to be an important factor in IVD disorders [37,38,39].

## 3. Current Treatment of IVDD

The current treatments for IVDD range from physiotherapy and medication to surgically invasive interventions [40,41]. If the degeneration is not severe, a healthy lifestyle, proper and regular exercise, weight loss, and physical therapy are crucial and simple ways of treating IVDD. Likewise, non-steroidal anti-inflammatory drugs help to control the pain. However, they may have limited effect on pain relief and cause side effects [42]. Surgical interventions include partial resection of the herniated disc and discectomy with fusion in which, after obliterating the disc completely, the intervertebral cage is inserted [43,44]. Back pain may decrease after spinal surgery, but there is increased risk of further degeneration at operated or adjacent levels and the pain often recurs [4]. Therefore, there is an urgent necessity for progress in more ideal and worthwhile biological treatments targeting the root cause of IVDD. Such treatments could be clinically applied to suppress the underlying disc degeneration, protecting discs from degeneration by promoting cell viability in the NP, maintaining homeostasis, and preserving the ECM. These may include injections of various growth factors such as growth differentiation factor-5 (GDF-5) with or without carriers, the use of cells such as MSCs, tissue engineering, and various genetic modifications through gene therapy [45,46].

## 4. Biological Therapies for IVDD

As surgical interventions involve excision of the pathological disc and spinal fusion, they result in loss of function, immobilization, and acceleration of additional complications. Thus, the development of novel biological therapies, including injection of growth factors, cell-based therapies, and gene therapies for disc degeneration, is of crucial importance in the present circumstances [47].

### 4.1. Growth Factor Based Therapy

In the IVD, homeostasis is maintained by the dynamic balance between catabolism and anabolism [4]. Numerous growth factors have shown the potential to modulate these effects in degenerated discs [47]. The direct injection of growth factors to regenerate the degenerated IVD is considered a favorable biological therapeutic modality [4]. Abundant studies in experimental models have explained cell proliferation and ECM synthesis in the IVD by growth factors such as basic fibroblast growth factor (bFGF or FGF2), epidermal growth factor (EGF), GDF-5, insulin-like growth factor-1 (IGF-1), osteogenic protein-1 (OP-1), and bone morphogenetic protein-2 (BMP-2). Similarly, by promoting the synthesis of the NP and inner annulus in a degenerated disc, exogenous GDF-5, BMP-2, and transforming growth factor-β (TGF-β) repaired inner annular chondrocytes. However, IGF-1 and bFGF showed weak effects on these cells [48,49,50,51,52]. Pro-inflammatory cytokines, as well as proteolytic enzymes, such as disintegrin and metalloproteinase with thrombospondin motifs (ADAMPTS) and matrix metalloproteinases (MMPs), carry out a crucial role in the ECM degradation of the NP. Thus, the transplantation of specific antagonists of these enzymes into the IVD has been scrutinized as a possible method to suppress further degradation of ECM [53]. The single injection of growth factors has a narrow effect because of their short half-lives (only hours to days), interstitial solubility, the absence of a suitable carrier, and an extremely small number of viable cells to be activated in the degenerated IVD. Furthermore, repeated injections of these growth factors may cause inflammatory reactions and additional injuries. Therefore, growth factor-based therapy has faced substantial obstacles in the clinical setting [54].

### 4.2. Gene Therapy

Gene therapy approaches exert long-lasting effects, thereby overcoming the short half-life of biological factors injected singly. Gene therapy transfers one or more therapeutic genes into the target cells by using vectors (viral and non-viral) to modify the functioning state of recipient cells by releasing the content of the donor gene [4]. Research on the use of gene therapy for IVD regeneration, has focused on therapeutic genes encoding anabolic factors, such as BMPs, GDF-5, LIM mineralization protein (LMP-1); transcription factors, such as SBY-box transcription factor-9 (SOX-9); anti-inflammatory agents, such as interleukin 1 receptor antagonist (IL1ra); and metalloproteinase inhibitors, such as tissue inhibitor of metalloproteinase (TIMP-1). Viral and non-viral vectors are the two main gene delivery methods [29]. Numerous studies have demonstrated methods for transferring a gene of interest into IVD cells. TGF-β, BMP-2, LMP-1, chondroitinase ABC, and TIMP are examples of exogenous gene products that can promote in the production of ECM in the IVD [55,56]. Even though viral vectors are the vehicles of choice for providing efficient gene delivery, safety concerns should be taken into consideration, as misdirected injections may cause hazardous effects on the surrounding tissue and uncontrolled gene expression. Additionally, significant expenses are involved in the manufacturing of vectors for gene therapy [4,57].

## 5. Mesenchymal Stem Cell Therapy in IVDD

MSCs are pluripotent non-hematopoietic stem cells acquired from adipose tissue, the umbilical cord, bone marrow, placenta, amniotic fluid, fat, dental pulp, and other sources [58,59]. When cultured in vitro, they have the ability to differentiate into adipocytes, osteoblasts, and chondroblasts [60]. Since MSCs do not express major histocompatibility complex-II (MHC-II) and costimulatory molecules such as CD 86, CD40, and CD80, they are considered immune-privileged cells [61]. MSCs, which have high regenerative, anti-inflammatory, anti-apoptotic, anti-angiogenic, anti-fibrotic and immunomodulatory abilities, can repair tissue damage and regulate cellular immunity by producing various types of bioactive molecules, such as platelet-derived growth factor (PDGF), matrix metalloproteinase-9 (MMP-9), matrix metalloproteinase-2 (MMP-2), interleukin-6 (IL-6), and insulin-like growth factor-1 (IGF-1) when they come into contact with a suitable host [62,63,64]. In the present scenario, MSCs are considered safe and show drastic therapeutic relevance in the sector of regenerative medicine [65]. About 60 clinical trials on MSCs have been approved by the US FDA, with focuses including hematopoietic stem cell transplantation, autoimmune diseases, vectors of gene therapy, and repair of damaged tissue [66]. However, worldwide, about 1000 clinical trials using MSCs to treat various diseases have been announced [67]. In light of the promising results of using MSCs in different disease conditions, studies on MSCs in treating IVDD have emerged as the center of attraction for many researchers. In recent decades, numerous preclinical studies in different animal models have investigated the treatment of IVDD using different types of MSCs [68]. MSC transplantation in the degenerated environment of the IVD shows promising results by restoring homeostasis in the disc. In addition to this, MSCs have the capacity to differentiate into NP-like cells and exert immunomodulatory and anti-catabolic effects [69]. Interestingly, recent studies reported that MSCs and NPCs engage in crosstalk through trophic factors to protect and proliferate NPCs in the degenerated disc by proliferating ECM synthesis, downregulating pro-inflammatory cytokines, and up-regulating the gene expression of NP markers such as SOX9, type II collagen, and proteoglycans [70]. Similarly, the role of MSC paracrine activity in promoting IVD regeneration has been investigated in several preclinical studies [68,71,72,73]. Increased mRNA expression of ECM genes s SOX9, collagen type 2 (COL2A1), and aggrecan (ACAN) and downregulation of MMPs, ADAMPTS, and pro-inflammatory factors were observed in comparison to monocultures when paracrine interactions in a co-culture system were investigated after obtaining MSCs, AF, and NPCs from same donor [74]. Likewise, an increase in the proliferation of NPCs, a higher yield of type II collagen, downregulation of MMP-9 expression, and decreased TGF-βand nuclear factor-κB (NF-κB) signaling in pre-senescent NPCs were observed in a co-culture system that used bone marrow mesenchymal stem cells (BM-MSCs) for TNF-α-induced NPC degradation [75]. To analyze the feasibility and potential clinical efficacy of the implantation of autologous MSCs embedded in tricalcium phosphate as a therapeutic alternative to bone grafting in patients with degenerative disc disease (DDD), a phase I/II clinical trial was designed; the results revealed that 80% of the patients underwent lumbar fusion with no adverse side issues [76]. Similarly, in a pilot study conducted on 10 patients with IVDD, autologous BM-MSCs were incorporated in the NP region without a cell carrier, which revealed an improvement in LBP and disability at 12 months of injection. In addition to this, an increase in water content was observed on T2-weighted MRI, whereas restoration of disc height was not noticeable [8]. The same results were seen in two female patients with IVDD 2 years after the injection of autologous BM-MSCs with 20 collagen porous sponges [77].

Even though MSC treatment has tremendous utility in the clinical and preclinical aspects, it has some limitations. Firstly, many studies concluded that transplanted cells could not survive in the hypoxic environment of the disc. Likewise, various inflammatory mediators, the low pH, low glucose levels, and hyperosmolarity present in the degenerated disc may disturb the function of transplanted MSCs [78]. Secondly, such therapies may pose a risk of granulocytosis, graft rejection, ectopic bone formation, and microvascular embolism [62,79]. However, in a pilot study using autologous bone marrow stem cell injection, researchers obtained significant recovery of pain and disability in 3 months [8]. Similarly, in another study, five patients with degenerative disc disease received an intra-discal transplantation of autologous BM-MSCs. The data showed clinical improvement at 4–6 years of follow up [10]. In the study performed by Noriega et al., in 24 patients with IVDD, using allogenic BM-MSCs, the effects of the injected cells were nearly completed at 3 months and were maintained at 6 and 12 months. Further, MRI results also demonstrated partial disc healing. Further, in one study, the researchers demonstrated that injection of allogenic hypoxic-cultured MSCs was more effective for disc height maintenance [80]. Finally, a standardized protocol must be formulated and studied for culturing procedures, the method of administering treatment, better timing, and the selection of suitable donor cells in the future [57,81,82]. In addition to this, studies should be done to determine the pH level and appropriate cell type transplantation, the type of carrier transplantation, the behavior of MSCs in vivo, and possible factors that may induce MSCs to differentiate into either NP-like cells or annular cells. Eventually, human trials will be obligatory to guide the safety, feasibility, and potentiality of MSC transplantation [83]. Thus, novel therapeutic methods are emerging, including the generation of cell-free techniques. Therefore, various stem cells, including MSCs, can generate exosomes with equivalent therapeutic effects. Further, they also contain various paracrine mediators, which are considered as a potential cell-free therapy for IVDD [57]. Exosomes derived from MSCs contain of a variety of regulatory factors that inhibit apoptosis, inhibit ECM degradation, downregulate inflammation, and promote chondrogenic differentiation, which could prevent IVDD [84] (Table 1 and Table 2).

## 6. Mesenchymal Stem Cell-Derived Exosomes

### 6.1. Exosome Biogenesis

Exosomes, which belong to the category of extracellular vesicles (EVs), are membrane-bound and produced by nearly all cells, tissues, and body fluids, including semen, plasma, urine, blood, saliva, breast milk, amniotic fluid, ascites fluid, cerebrospinal fluid, and bile [93]. Exosomes are categorized as EVs with a size range of 30–100 nm, secreted by multivesicular bodies (MVBs) in the endosomal pathway, microvesicles with a size of 100–1000 nm obtained from the outward budding of the plasma membrane, and apoptotic bodies having a size of 1000–5000 nm, produced by cells in apoptotic conditions; these types differ in size, function and biological origin [57,94]. Exosomal biogenesis proceeds with early endosomes, which are derived from an endosomal compartment and liberated into the extracellular environment upon fusion of multivesicular bodies (MVBs) comprising intraluminal vesicles with the plasma membrane. The loading and biogenesis of exosomes are regulated by endosomal sorting complexes required for transport (ESCRT) loaded on MVBs. Tumor susceptibility gene 101 (Tsg101), vacuolar protein sorting-associated protein (Vps4), and Alix are the associated proteins that participate in these processes [95]. Exosomes derived from MSCs depend on external cues to act normally. Exosomes with different constitutions and contents are secreted under different stimuli from the same parental cells [96,97]. They are taken up by recipient cells with the help of the body’s circulatory system. The recipient cells commonly take up exosomes in three ways: ligand–receptor interaction, endocytosis by the recipient cell, and direct fusion with the membrane [62] (Figure 1). Thus, exosomes act as intercellular communicators by transmitting signals and transporting substances.

### 6.2. Mesenchymal Stem Cell-Derived Exosome Components 

Exosomes are considered a miniature form of the parental cell, and many constitutive elements have been recognized in exosomes from various cell types, including about 4400 proteins, 194 lipids, 1639 mRNAs, and 764 microRNAs (miRNAs), which determine the biological functions of exosomes [98]. With tremendous improvements in the field of biotechnology, the composition of exosomes has been extensively explored [99]. Lipids in exosomes are mainly located in the membrane, which includes phosphatidylserine, phosphatidic acids, cholesterol, sphingomyelin, arachidonic acid, prostaglandins, and leukotrienes. Lipids are not only engaged in maintaining biological stability, but also involved in the formation and release of exosomes [100]. Furthermore, exosomes abundantly contain varieties of proteins, such as cytoskeleton components, tetraspanins, heat shock proteins, and many more [101]. In addition to proteins and lipids, exosomes are extremely rich in various nucleic acids including genomic DNA, complementary DNA, mitochondrial DNA, long noncoding RNAs, circular RNAs, miRNAs, and mRNAs reflecting the mutational status of the source cells [99,102].

MSCs can generate a greater amount of exosomes than other cell types by the paracrine pathway [103]. Not only do MSC-derived exosomes express common surface such as CD9, CD81, CD63, Alix, and Tsg101, and heat shock proteins (HSP90, HSP70, and HSP60), but interestingly, they also express several adhesion molecules in their membranes such as CD73, CD44, and CD29 [104,105]. As with other exosomes, those derived from MSCs are also enhanced with nucleic acids, namely long non-coding RNA, miRNA, and mRNA [106]. According to various studies, miRNAs are a topic of interest as a class of endogenous noncoding RNAs with a length of about 20–25 nucleotides, that regulate gene expression, the immune response, and other physiological processes by acting on the behavior of recipient cells [107,108,109]. They resemble the key biomolecular player and are involved actively via different pathways in cell proliferation, immunomodulation, apoptosis, and ECM anabolism and catabolism [110]. Many miRNAs that have been identified in MSC-derived exosomes are MSC exosomal effector molecules. They are involved in numerous biological activities, including apoptosis, cell differentiation, angiogenesis, and inflammatory pathways. The miRNAs that have been identified in MSC-derived exosomes are tumor growth inhibitor miRNAs (miR-122, miR-125b, miR-24, miR-31 miR-223, miR-451, miR-214, and miR-23b) and inflammatory modulatory miRNAs (miR-146 and miR-155) [22,111]. These observations suggest that exosomes derived from MSCs have promising therapeutic potential by inhibiting inflammation and oxidative stress and promoting cell proliferation and ECM synthesis in various clinical conditions.

## 7. Application of Mesenchymal Stem Cell-Derived Exosomes in Different Tissues

Many studies on MSC-derived exosomes have shown excellent results in promoting tissue regeneration and homeostasis. In one study, exosomes derived from BM-MSCs promoted bone and vascular regeneration in a rat femur nonunion model [112]. Similarly, vascularization of endothelial cells was promoted by using exosomes derived from adipose stem cells [113]. Likewise, exosomes derived from human MSCs also showed significant results in repairing jaw joints and preserving the ECM in the tissue [114]. MSC exosomes derived from the human umbilical cord also help in the promotion of angiogenesis and repairing burn wounds on the skin [115]. MSC-derived exosomes, as small transporters, can easily cross the blood–brain barrier (BBB), which has led to considerable interest in their use to treat various degenerative brain disorders, such as Parkinson’s disease, Alzheimer’s disease, stroke, and autism [116,117]. Likewise, the MSC-derived exosomes carrying miRNAs can slow the progression of Alzheimer’s and Parkinson’s diseases by mediating oxidative stress [118]. The MSC-derived exosomes and embryonic stem cell-derived exosomes also enhance neovascularization, improve cardiomyocyte survival, and reduce fibrosis to ameliorate cardiac function in the hearts of patients following myocardial infarction [119]. MSC-derived exosomes also have promising therapeutic benefits in the proliferation of skeletal muscle regeneration [120]. Thus, these findings suggest that MSCs-derived exosomes have an adequate capability for cell proliferation, angiogenesis, maintenance of homeostasis, induction of ECM, and inhibition of inflammation and oxidative stress. 

## 8. Effects of Mesenchymal Stem Cell-Derived Exosomes on IVDD

Tremendous progress has been seen in the use of MSC transplantation to treat IVDD. Furthermore, various in vitro and in vivo studies have demonstrated many beneficial roles of MSC-derived exosomes for cell-based therapies [121]. From these various findings, the major pathological hallmarks of IVDD include reduced ECM, substantial cell loss, increased oxidative stress, and inflammation in the IVD. Interestingly, MSC-derived exosomes may restore the degenerated IVD in the normal form to some extent by decreasing those pathological markers. However, due to the complex structure and harsh nature of the IVD, there are numerous difficulties in utilizing stem cell approaches to treat IVDD [122]. The specific mechanisms of MSC-derived exosomes in IVD are described below. Table 3 shows various preclinical studies of MSC-derived exosomes in IVDD. Figure 2 shows the administration of MSC-derived exosomes with their main therapeutic mechanism and various effects in IVDD.

### 8.1. Maintenance of Homeostasis of the Extracellular Matrix

Imbalances in ECM homeostasis in the IVD play a vital role in IVDD [134]. Many studies have shown that when the IVD degenerates, the IVD cells may produce certain pro-inflammatory cytokines such as interleukin-1β, TNF-α, and other infiltrating immune cells. As a result, the upregulation of catabolic enzymes such as ADAMTS and MMPs and the downregulation of ECM synthesis-related molecules, (namely, aggrecan, type II collagen, and proteoglycan) occur in the disc. This phenomenon creates an imbalance in homeostasis in the ECM [30,36].

Abundant data have revealed that MSC-derived exosomes maintain the ECM homeostasis by increasing the expression of ECM-related molecules [15,73,82,135,136,137,138,139,140,141]. An in-vitro study performed with human MSCs resulted in a 50% increment in the proliferation of NPCs and a decrease in cellular apoptosis and inflammation in degenerated discs. Likewise, excessive ECM production was also seen as soon as day 7. In addition to this, three-times-higher production of ECM was observed in degenerated disc cells treated with MSC-derived exosomes than in control cultures, and small extracellular vesicle (sEV) treatment also suppressed the secretion of the inflammatory cytokine MMP-1 [133]. Likewise, NPC-derived exosomes both promoted BM-MSC migration and induced BM-MSCs to differentiate into the NP-phenotype. Similarly, human BM-MSC-derived exosomes promoted NPC proliferation and healthier ECM production in the degenerated NPCs by upregulating anabolic matrix-protective genes, such as aggrecan, type II collagen, and SOX-9, and downregulating matrix-degrading genes such as MMP-1 and MMP-3. These findings were indicative of stabilization of the equilibrium between catabolism and anabolism in the IVD [18]. In a rat model of IVD, when MSC-derived exosomes were injected intradiscally, they improved NPC senescence, increased the disc height, improved IVDD, and recovered the age-related function by triggering the sirtuin 6 pathway by downregulating the level of cyclic adenosine monophosphate (cAMP) specific hydrolase phosphodiesterase 4D (PDE4D) through delivering miR-105-5p in-vitro. Furthermore, it was found that sEVs from induced pluripotent stem cell-derived mesenchymal stem cells downregulated the markers of catabolism (ADAMPIS-4 and MMP-3) and upregulated anabolic markers (aggrecan and collagen) [126]. The exosomal miR-223 downregulated lipopolysaccharides induced NPCs, upregulated the proliferation of ECM-related genes such as aggrecan and type II collagen, and downregulated matrix-degrading enzymes (e.g., ADAMPTS4, MMP3, and MMP13) and expression of NF-κB signaling-related proteins [142]. Likewise, NPCs-derived exosomes both promoted BM-MSCs migration and induced them to differentiate into NP-like phenotypes. Similarly, BM-MSC-derived exosomes promoted NPC proliferation and healthier ECM production in degenerated NPCs [18]. Likewise, exosomes derived from adipose-derived MSCs downregulated MMP-13 while increasing type II collagen expression, thereby promoting ECM formation in degenerated IVDs [143]. In several studies, the effects of MSC-derived exosomes in IVDs with biochemical challenges, such as IL-1β, TNF-α, advanced glycation end products, high glucose levels, tert-butyl hydroxide, and acidic pH were incorporated to stimulate ECM synthesis. Interestingly, they showed promising results in NPCs by upregulating aggrecan and type II collagen and downregulating matrix-degrading enzymes [15,19,20,73,123,124,125,129,130,132,144,145].

### 8.2. Inhibition of Apoptosis

The degenerated disc is characterized by low glucose, high osmotic pressure, low pH, hypoxia, increased mechanical loading, and increased osmotic pressure; as a consequence, cell viability is reduced and apoptosis increases [6]. Apoptosis, which leads to a decrease in the number of viable cells, is a major event and is thought to play a vital role in in the IVDD. Thus, minimizing cell death has been explored as an important therapeutic approach for IVDD [146]. Many studies have concluded that exosomes derived from MSCs promote the proliferation of viable cells in IVD and inhibit the apoptosis mechanism [147]. 

One of the root causes of the progression of IVDD is excessive apoptosis in NPCs [130]. Previous research has shown that when MSC-derived exosomes were delivered in a rat model of IDD, NPCs were prevented from undergoing the apoptotic process and IVDD was alleviated via miR-21 contained in exosomes, as exosomal miR-21 inhibits phosphatase and tensin homolog (PTEN); therefore, activation of the phosphoinositide 3-kinase/protein kinase B (PI3K/Akt) pathway occurred in apoptotic NPCs, and this phenomenon decreased the degree of activation level of downstream factors of Bad, Bax, and caspase-3 and ultimately hindered TNF-α mediated apoptosis [20]. The exosomes derived from BM-MSCs prevent NPC apoptosis and degradation, inhibit fibrotic deposition and ECM disintegration, and obstruct the IVD being degenerated via miR-532-5p, as the miR-532-5p gene is directed by Ras association domain-containing protein 5 (RASSF5) to alleviate the progression of IVDD [130]. Similarly, BM-MSC-derived exosomes relieved NPC apoptosis and stimulated NPC proliferation by reducing IL-1β-induced inflammatory cytokines secretion and mitogen-activated protein kinase (MAPK) signaling activation. Additionally, MSCs-derived exosomes inhibited NPCs apoptosis and MAPK signaling by delivering miR-142-3p, which targets mixed-lineage kinase 3 (MLK3) [132]. Likewise, a study on exosomes derived from human placental mesenchymal stromal cells with antagomiR-4450 enhanced the attenuation of apoptosis and inflammation and elevated cell migration and proliferation partially via increasing levels of zinc finger protein 121 (ZNF121) [124]. In one study, the proliferation of NPCs and inhibition of cell apoptosis were observed after treatment with miR-199a from MSC-derived exosomes by downregulating GREM1 in vitro [147]. Exosomes from MSCs could diminish endoplasmic reticulum (ER) stress-induced apoptosis by activating the Akt and extracellular signal regulated kinase (ERK) signaling pathway. Similarly, it was found that delivery of MSC-derived exosomes in vivo modulated ER-stress-related apoptosis and delayed IDD progression, and favored IVD regeneration in a rat tail model [19]. Likewise, when exosomes from MSCs were co-incubated with advanced glycation end products, NPCs apoptosis was inhibited by downregulating caspase-3 and caspase-12 [19]. A study performed on BM-MSC-derived exosomes revealed that BM-MSC-derived exosomes could promote proliferation and reduce the apoptosis induced by IL-1 β in AF cells by activating PI3K/AKT/mTOR signaling pathway-mediated autophagy. Furthermore, BM-MSC-derived exosomes suppressed the IL-1β-induced inflammation in AF cells of IVD [129]. A study found that sub-endplate injection of MSC-derived exosomes ameliorated IVDD in rats by reducing ER-stress related apoptosis and calcification in endplate chondrocytes (EPCs) under oxidative stress induced by tert-butyl hyperoxide. It was found that MSCs- exosomes were rich in miRNA-31p, and downregulation of MiRNA-31p inhibited exosomal protective benefits in the cartilaginous endplate [125].

### 8.3. Anti-Oxidation Effects

One of the major factors in IVDD is excessive oxidative stress which encourages the expression of catabolic factors in the disc [147,148]. During degeneration, the homeostasis between the generation of oxygen free radicals and antioxidant defense is disrupted. As a result, formation of reactive oxygen species (ROS), oxidative stress, and finally, mitochondrial apoptosis in NPCs take place. Therefore, the oxidative stress mechanism is a crucial topic for IVDD [57]. In a study, researchers demonstrated that mitochondrial proteins could be supplied to NPCs by MSC-derived exosomes, which enable the regeneration of the destroyed mitochondrial components. They also presented an analysis of the proteomic database that MSC-derived exosomes contained antioxidant proteins, including peroxiredoxin-1, and glutathione peroxidase 4 [123]. In a rat model of IVDD, inhibition of apoptosis and calcification in EPCs was observed by targeting activating the transcription factor 6 (ATF6)-related ER stress pathway through the delivery of miR-31-5p [125].

### 8.4. Anti-Inflammatory Activity 

Various pro-inflammatory cytokines such as chemokine (C-C motif) ligands 545 (CCL545), chemokine (C-C motif) ligands 4 (CCL4), chemokine (C-C motif) ligands 3 (CCL3), chemokine (C-X-C motif) ligand 1 (CXCL1), and chemokine (C motif) ligand (XCL1) are produced by degenerative disc cells themselves during the process of IVDD, these cytokines attract other leukocytes, resulting in the production of pro-inflammatory cytokines such as TNF-α, IL-1, IL-2, IL-6, and IL-8, which upregulate the matrix catabolic enzymes, causing IVDD to take place quickly [149]. Triggering of these cytokines give rise to a series of pathologic responses in the IVD, causing autophagy, senescence, and apoptosis [64]. Exosomes from MSCs played an anti-inflammatory role in treating pathological NPCs by suppressing inflammatory mediators and NLRP3 inflammasome activation in the rabbit model of IVDD. Furthermore, exosomes might supply mitochondrial proteins to NPCs, so that defected mitochondria could be replaced with normal functions [123]. The exosomal miR-223 from plasma might play a homeostatic role in downregulating inflammatory responses through intracellular communication [150]. The BM-MSC-derived exosomes relieved NPC apoptosis by reducing IL-1β induced inflammatory cytokines secretion and MAPK signaling activation. Additionally, MSC-derived exosomes inhibited NPC apoptosis and MAPK signaling by delivering miR-142-3p, which targets MLK3 [132]. Likewise, a reduction in the inflammatory response in AF cells was seen when cells were treated with BM-MSC-derived exosomes by inhibiting PI3K/AKT/mTOR signaling pathway mediated autophagy [129].

## 9. Discussion

Exosomes are membranous nano-sized vesicles that are secreted by a variety of cells; exosomes merge their membrane contents into the recipient cell membrane and transfer factors into recipient cells. MSC-derived exosomes bear a compound load of nucleic acids, lipids, and proteins, including abundant miRNAs, and pass on these contents into the recipient cells, supporting the potential role of MSC-derived exosomes as an alternative, cell-free therapy. Thus, these exosome-delivered effectors, particularly miRNAs, can supply an effective therapeutic method of attenuating the recipient NP cell apoptosis, promoting cell proliferation, and promoting the production of ECM [21]. Despite the tremendous applications of MSC-derived exosomes, most studies have been based on IVDD models in rodents not in humans [151]. This is an issue because puncture or drug-induced IVDD in animal models may not fully stimulate the effects of MSC-derived exosomes on human IVDD.

Even though MSCs are considered a currently trending topic in the field of regenerative medicine, their clinical application is challenging due to storage limitations and cell senescence that may occur during in vitro expansion. Exosomes, through paracrine signaling, are secreted by MSCs, and they have similar properties to those of MSCs. MSC-derived exosomes have low immunogenicity, low cytotoxicity, and high repairability. In addition to this, they are of nano-scale size and can be stored conveniently. Thus, their clinical applications are expanding tremendously [152,153]. Exosomes originating from MSCs have gained popularity in the field of regenerative medicine due to their potential ability to generate damaged tissue [29]. Likewise, many studies have demonstrated that MSC-derived exosomes are a superior and convenient substitute for MSCs [152,154,155,156,157,158,159,160]. Likewise, exosomes can be merged easily with the existing compositions or carriers to deliver drugs. Exosomes can target specific cell types or tissues after being engineered very precisely. They can avoid the side effects of cell therapy as they are cell-free, and they are safer with no risk of tumor formation, whereas previous studies have demonstrated the possibility of tumorigenicity in MSC-based therapies [156,157]. MSC-derived exosomes are more stable to store and cost-effective than MSCs [62]. Exosomes may be combined with some therapeutic cargo to obtain promising results [158]. Injectable hydrogels are also used as carriers of therapeutic agents such as bioactive molecules in treating and repairing tissue regeneration [159]. In one study, the use of a photoinduced imine crosslinking hydrogel glue with stem cell-derived exosomes showed an excellent effect on cartilage regeneration. Exosomes deliver drugs to their target sites because of their natural targeting competence, and immense ability to permeate biological barriers. The bioactive substances that are delivered by exosomes are easily degraded via multiple pathways [64]. Therefore, interest in the use of exosomes for disc regeneration is increasing [160]. In the present scenario, for treating IVDD, only one clinical trial (phase 1) is being conducted in India; this trial involves using intradiscal injections of platelet-rich plasma that is enriched with exosomes in 30 participants [161]. Despite being a promising way of treating IVDD, various drawbacks and challenges still exist. We confront many challenges that need to be solved before using MSC-derived exosomes in clinical trials. The first one to be noted is a selection of suitable MSC donors that generate appropriate exosomes for a specific tissue. This is challenging because disc degeneration is a completely complex and incompletely understood multifactorial process with contributions from mechanical stresses, genetic factors, cellular senescence, alterations in nutrition via a narrow vascular supply, or natural changes [162]. Furthermore, the IVD is the largest avascular tissue in the body, having a complex pathological environment [163]. Thus, it is very difficult to select the appropriate exosomes that are derived from stem cells and clarify their specific actions. 

Secondly, we need to develop unified worldwide standard protocols for exosome isolation, separation, purification, and profiling that meet the standards of good manufacturing practice [162]. In addition to this, the storage and transportation of exosomes is an important issue that should be taken seriously. A study demonstrated that freeze-drying facilitated the transportation and storage of exosomes as well as some degree of purification [145]. However, more studies should be carried out to identify the exact mechanism behind these findings to achieve a pure yield of exosomes. Thirdly, the pharmacological characteristics of exosomes in vivo, such as their bio-distribution, bioavailability, and pharmacokinetics, have still not been studied [164]. The exact mechanism of action of exosomes should be well formulated. The fourth challenge is the need to study the exact route of administration, dosing, dose toxicity, targeting and interval of exosomes, which is still a major problem in the field of regenerative medicine [165]. While considering this, we should not forget the optimization of culture conditions for exosomes [166]. To establish the role of MSC-derived exosomes in treating IVDD, the production of safer, more advanced functionalized exosomes and culture methods are important issues [57]. Lastly, it is mandatory to monitor and ensure the safety of the patient population, which will need to be addressed and acted on properly before setting up clinical trials [161]. 

## 10. Conclusions and Future Perspectives

In conclusion, MSC-derived exosomes have diverse prospects for repairing degenerated IVDs. They can promote the proliferation of NPCs, maintain the homeostasis of the ECM, inhibit cell apoptosis, and downregulate inflammation in the IVD. Despite their benefits, their detailed mechanisms are still unknown. Thus, further innovative investigations are needed to elucidate the exact mechanism, isolation, storage, dosage, mode of administration, and safety of MSC-derived exosomes in successful clinical settings for treating IVD.

Exosomes can be obtained from any cell source according to the purpose at hand. The establishment of therapeutic strategies is vital in making exosomal therapy more effective and promising. MSC-derived exosomes are suitable candidates for repairing degenerated discs by maintaining homeostasis. To obtain the desired therapeutic effects, they can also be engineered and loaded with biomaterials to control the release in the targeted tissue [167]. Even though many previous results have demonstrated the promising therapeutic potential of MSC-derived exosomes, special attention should be given before using them in clinical trials [168]. The in vivo studies currently utilize only a single dose; however, if systemic injections were to be implemented to achieve the perfect therapeutic effects, multiple doses may be mandatory. Thus, in such situations, the safety and potency of multiple infusions and appropriate doses should be kept a top priority [161]. It would be necessary to develop assays capable of predicting the therapeutic potential of MSC-derived exosomes with high clinical sensitivity and specificity [169]. Furthermore, it is well known that exosomes originate from stem cells, but are not cells, which might lead to challenges in terms of defining their legal classifications and gaining approval from each country’s regulatory authorities for use in clinical trials [170]. In a recent study, exosome-mimetic nanovesicles (NVs) were formulated, as an alternative to natural exosomes derived from MSCs. NVs can be produced in higher yields; however, there is still a lack of an effective generation method for NVs [171]. Thus, MSC-derived exosomes should be inspected well for safety issues before use in clinical trials.

## Figures and Tables

**Figure 1 ijms-23-07306-f001:**
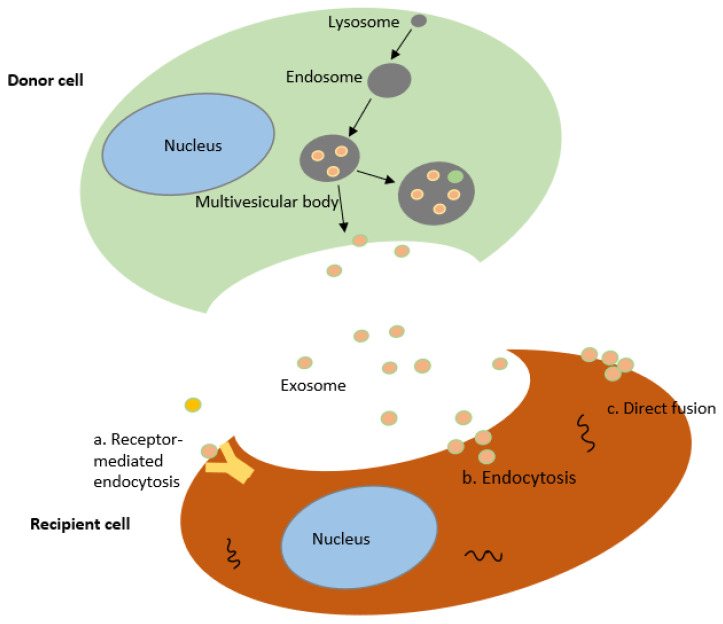
The biogenesis of exosomes. Exosomes are secreted by donor cells into the intercellular microenvironment via multivesicular bodies. Exosomes can transfer biologically functional molecules to recipient cells in three ways (a) intercellular signaling through receptor-ligand binding, (b) endocytosis of recipient cells, (c) direct fusion of exosomes with the recipient cells membrane, after which exosomes release their contents.

**Figure 2 ijms-23-07306-f002:**
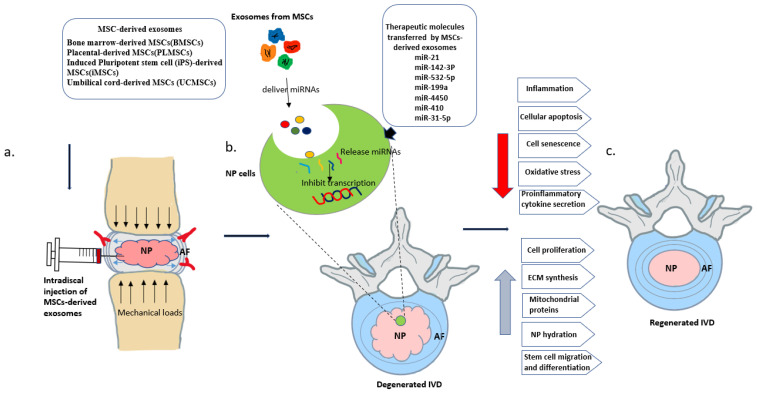
(**a**) General diagram of experimental intradiscal injection of MSC-derived exosomes in a degenerated IVD. (**b**) In NP cells of the degenerated IVD, therapeutic molecules (miRNAs) are transferred by MSC-derived exosomes that inhibit the translation process, thereby regulating multiple intracellular processes, including cell proliferation, apoptosis, and cytokine release. (**c**) Regeneration of IVD after MSC-derived exosomes treatment with an increase in cell proliferation, ECM synthesis, and NP hydration, and upregulation of mitochondrial proteins.

**Table 1 ijms-23-07306-t001:** Biological therapies in different studies with their effects.

Biological Therapies	Approach	Effects	References
Growth factor			
TGF-β1	In vitro	- Increase in NP cell proliferation and ECM synthesis, decreased expression of ADAMPTS-4 and -5	[85]
IGF-1	In vitro	- Increase in ECM synthesis, proteoglycan synthesis	[51]
IL-1β	In vitro	- Inhibited NP cell proliferation, increase expression of ADAMPTS-4 and -5	[85]
TNF-α and IL-1β	In vitro	- Increase in MMP-1	[50]
OP-1	In vivo	- Increased disc height and proteoglycan content in NP cells	[48]
GDF-5	In vivo	- Structural and functional maintainace of disc	[86]
Cell therapy			
MSCs	In vivo	- Slowing in disc height loss, increase in T2 weighted signal intensity, increased GAGs	[87]
BM-MSCs	In vivo	- MSCs can migrate out of the nucleus an undesirable bone formation may occur	[88]
Articular chondrocytes	In vivo	- Increased proteoglycan content and collagen type II synthesis	[89]
BMMSCs and AD-MSCs	In vivo	- -MSCs differentiate into NP cells phenotype, preserved water, and disc height restored	[90]
Gene therapy			
SOX-9	In vivo	- Increased matrix synthesis, increased aggrecan and collagen II	[91]
TIMP-1	In vivo	- Less MRI and histological evidence of degeneration	[56]
IGF-1	In vivo	- Reversed apoptotic rate of NP cells	[92]

**Table 2 ijms-23-07306-t002:** Advantages and limitations of biological therapies.

Biological Therapies	Advantages	Limitations	References
Growth factor therapy	- Stimulate the proliferation of IVD cells and accumulation of ECM	- Short half-life, absence of suitable carrier, lack of stability	[54]
Cell therapy	- Well-characterized, histocompatibility and safe	- Growth of osteophytes - Cell leakages and disc infection - Genetic modification is unsafe - Tumorigenesis	[32]
Gene herapy	- Exert long-lasting effects	- Misdirected injections may cause hazardous effects on the surrounding tissue and uncontrolled gene expression - Expensive in making vectors	[4,57]

**Table 3 ijms-23-07306-t003:** Pre-clinical studies of stem-derived exosomes with their therapeutic potential for IVDD.

Aim of Experiment	Origin of Exosome	Animal Model	In Vitro Study Conclusion	In Vivo Study Conclusion	Study Types Performed	References
- To evaluate the therapeutic potential of MSCs and associated exosomes on NPC pyroptosis.	MSCs	Mouse	- MSC-derived exosomes suppress NLRP3 pathway, play an anti-pyroptosis role, and prevent IVDD.	- The degree of severity of IVDD is alleviated by MSC-derived exosomes and miR-410.	In vitro In vivo	[15]
- To investigate the restorative effects of MSC-derived exosomes on H2O2-induced NPC inflammation and mitochondrial dysfunction.	BMSCs	Rabbit	- BM-MSC-derived exosomes act as an anti-inflammatory in pathological NPCs by suppressing inflammatory mediators and activating NLRP3 inflammasome and cleavage of caspase-1 by H_2_O_2_. - Mitochondrial-related proteins are recovered.	- MSC-derived exosomes reduce the progression of IVDD by delaying ECM degradation in the IVD.	In vitro In vivo	[123]
- To explore the potential therapeutic role of hPLMSC-derived exosomes carrying AntagimiR-450.	hPLMSCs	Mouse	- hPLMS-derived exosomes carrying antagomiR-4450 prevents IVDD by promoting the proliferating NPCs - EV-derived antagomiR-4450 reduces the expression of MMP-13, IL-6, and IL-1β while upregulating aggrecan.	- The antagomiR-4450 suppresses miR-4450, increases ZNF121 expression and decreases pro-inflammatory factors in mouse model of IVDD.	In vitro In vivo	[124]
- To prove the inhibition of the apoptosis of NPCs by modulating ER- stress with the assistance of MSC-derived exosomes.	BMSCs	Rat	- ER stress-induced NPC apoptosis is reduced by activating AKT and ERK signaling.	- ER stress-related apoptosis is modulated and degeneration is decreased in a rat tail model of IVDD.	In vitro In vivo	[19]
- To investigate the role of MSC-derived exosomes on EPCs by inhibiting apoptosis and calcification.	MSCs	Rat	- Reduction of apoptosis and calcification by MSC-derived exosome in EPCs are achieved by regulating the miR-31P/ATF6/ER stress pathway.	- Injections of MSC-derived exosomes in sub-endplate slow the progression of IVDD by downregulating miR-31-5p.	In vitro In vivo	[125]
- To explore the therapeutic potential of sEVs derived from iMSCs on IVDD.	iMSCs	Rat	- miR-105-5p contained in MSCs-exosomes restores anabolic markers such as aggrecan and collagen and lowers catabolic markers such as MMP-3 and ADAMPTS of the ECM and activates the SIRT6 pathway to restore age-related function.	- Intradiscal injections of MSC-derived exosomes downregulate the progression of IVDD by decreasing NPC senescence and restoring IVD height.	In vitro In vivo	[126]
- To study NPC apoptosis by normal cartilage endplate stem cell (CESC)-derived exosomes (N-Exos) in comparison with degenerated CEPC-derived exosomes.	CESCs	Rat	- CESCs diminish the apoptosis rate of NPCs in vitro	- NPC apoptosis is inhibited by N-Exos by activating PI3K/p-AKT/autophagy signaling pathway.	In vitro In vivo	[127]
- To explore the effect of UC-MSC-conditioned medium on high-glucose (HG)-induced degradation of NPMSCs produced ECM	UC-MSCs	-	- MSC-derived exosomes have the capacity to alleviate HG-induced ECM degradation through the p38 MAPK pathway. HG inhibits type II collagen and aggrecan expression in NPMSCs and gives rise to NPMSC apoptosis.	-	In vitro	[73]
- To detect the capacity of MSC-derived exosomes to attenuate apoptosis of NPCs elicited by pro-inflammatory cytokines	BM-MSCs	-	- MSC-derived exosomes enhance the expression of ECM, type II collagen, aggrecan, and reduce matrix degradation by downregulating matrix degrading enzymes, protecting NPCs from acidic pH-induced apoptosis in a pathologically acidic environment.	-	In vitro	[128]
- To study BM-MSC-derived exosomes that inhibited IL-1βinduced inflammation and apoptosis of AF	BM-MSCs	-	- BM-MSC-derived exosomes suppress IL-1β-induced inflammation and apoptosis and promote the cell proliferation of AF cells by activating the PI3K/AKT/mTOR signaling pathway.	-	In vitro	[129]
- To detect the inhibitory role of BM-MSC-derived exosomes in apoptotic, ECM degradation, and fibrosis deposition in TNF-α-induced NPCs	BM-MSCs	-	- BMC-MSC-derived exosomes lead to the inhibition of TNF-α-induced apoptotic process, imbalance of anabolism and catabolism levels, and proliferation of type I collagen in NPCs via the delivery of miR-532-5p.	-	In vitro	[130]
- To study the role of MSC-derived exosomes in protecting against IVDD by causing the proliferation of NPCs and inhibiting NPC apoptosis	BM-MSCs	Mouse	- Increase in the expressions of type II collagen and aggrecan are observed, while the apoptosis rate of NPCs is decreased after exosomes mediation. - miR-199a from MSCs exosome assist in repairing by targeting GREM1 and reducing the TGF-β pathway.	- Treating degenerated discs in mice with MSC-derived exosomes downregulates MMP-2, MMP-6, and TIMP1, while miR-199a levels are increased.	In vitro In vivo	[131]
- To detect the therapeutic potential of MSCs-derived exosomes in ameliorating NPCs apoptosis via delivering miR-142-3p	BM-MSCs	-	- MSC-derived exosomes relieve NPCs apoptosis by diminishing IL-1β-induced inflammatory secretion and activation of MAPK signaling - MSC-derived exosomes hinder NPC apoptosis and MAPK signaling by transporting miR-142-3p which targets MLK3.	-	In vitro	[132]
- To explore the protective effect of MSC-derived exosomes on NPC apoptosis and IVD degeneration and investigate the regulatory effect of miRNAs in MSC-exosomes and the associated mechanism of NPC apoptosis	BM-MSCs	Rat	- Delivery of miR-21 in MSC-derived exosomes alleviates TNF-α-induced NPC apoptosis by targeting PTEN via the PI3K-AKT pathway.	- Injections of MSC-derived exosomes intradiscally prevent the apoptosis of NPC and protect the disc from degeneration.	In vitro In vivo	[20]
- To investigate the effect of human MSC-derived exosome on degenerated disc cells in 3D in vitro model	BM-MSCs	-	- Treatment with MSC-derived exosomes enhances cell viability and proliferation, promotes the production of ECM components, and suppresses apoptosis in degenerated disc cells.	-	In vitro	[133]

## Data Availability

Not Applicable.

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
