# Peer review of "Mesenchymal Stem Cell-Derived Exosomes and Intervertebral Disc Regeneration: Review"

_ijms, 2022, doi:10.3390/ijms23137306_

Round 1
Reviewer 1 Report
Title might be better as Mesenchymal Stem Cell-Derived Exosomes and Intervertebral Disc Regeneration: Review
Abstract: remove the word "recently" from the sentence with the word "decades"
Introduction: "bed rest" is not a treatment for disc degeneration or low back pain except for fractures or very briefly for acute pain. I would remove "bed rest" and rewrite the phrase as follows: "physical therapy, pharmacologic interventions and surgery are the main clinical treatments..."
The sentence starting with "Therefore" can be deleted. The sentence starting with "To overcome" could start as follows: "Various new approaches such as growth factors, biologics, gene transfection...." followed by reference [4]. The sentence starting with "However" can be deleted. The last sentence of the Introduction can start "For this review, we searched...."
2. Pathophysiology - "unsanitary lifestyle" - this is vague and disc infection as a common cause of IVDD is not proven. Perhaps you can insert the phrase "a diet leading to obesity" instead.
3) Current treatment - "but eventually, the degeneration.." can be changed to "but there is increased risk of further degeneration at operated or adjacent levels and the pain often recurs."
4.1 line 134 delete the extra "the"
5. Mesenchymal "several preclinical studies" should be followed by several references, not one reference [71].
Line 209 - did you mean "pre-senescent NPCs"?
Line 226 - there is no risk of graft rejection with autologous cells, so you need to specify this is true for "allogeneic" cells.
6.2 Mesenchymal - line 290-291 delete the phrase "of exosomes" and change "Similar in normal exosomes..." to "As with other exosomes, those derived from MSCs are also enhanced with..."
7. Application "vascularization of endothelial cells" - what does that mean?
Line 315 "human stem cells" - do you mean "human MSCs" ?
Line 330 - the sentence "Extensive research..." is a conclusion and not an introduction to the next section so just delete it.
Line 355 You start the sentence with "Abundant data" but end it with only two references! If the data were really abundant, then you need at least 3 or more references, not two.
Line 364, the sentence "Similarly, " should state if these are human BM-MSCs as in "Similarly, human BM-MSCs... " if that is true.
Line 367 - the sentence should end after MMP-3, followed by "These findings..."
Figure 2a - It should be made clear that this is a general diagram, not one related to a specific paper. Thus consider: "Figure 2. General diagram of experimental intradiscal injection of MSC-derived....."
Discussion: line 504 "Likewise, 'many' studies..." should be followed by 'many' references, not just Ref [139]!
Line 527 - "unsuitable environment for NPC proliferation" yet NPCs are naturally there! Isn't that like saying that a desert is an unsuitable environment for a cactus?
In general, the paper needs some editing, not necessarily from an English language expert, but from an expert in language and literature to reduce unnecessary words, phrases and sentences and make the paper more concise and better organized.
Author Response
Response to Reviewer 1 comments
#1 Title might be better as Mesenchymal Stem Cell-Derived Exosomes and Intervertebral Disc Regeneration: Review
Abstract: remove the word "recently" from the sentence with the word "decades"
Introduction: "bed rest" is not a treatment for disc degeneration or low back pain except for fractures or very briefly for acute pain. I would remove "bed rest" and rewrite the phrase as follows: "physical therapy, pharmacologic interventions and surgery are the main clinical treatments..."
The sentence starting with "Therefore" can be deleted. The sentence starting with "To overcome" could start as follows: "Various new approaches such as growth factors, biologics, gene transfection...." followed by reference [4]. The sentence starting with "However" can be deleted. The last sentence of the Introduction can start "For this review, we searched...."
Response #1:
We are very grateful for the reviewer’s insightful comments on our paper. We have changed the title according to the reviewer’s comments.
Thank you so much for the reviewer’s valuable suggestions. We have removed the word “recently” from the sentence with the word “decades”. (Edit line 19)
We are thankful for the reviewers’ suggestions. We have changed and rewritten it according to the reviewer’s suggestions. (Edit lines 36, 37)
We are thankful for the reviewers’ suggestions. We have deleted and rewritten it according to the reviewer’s suggestions. (Edit lines 39, 40, 43, 74, and 75)
#2. Pathophysiology - "unsanitary lifestyle" - this is vague and disc infection as a common cause of IVDD is not proven. Perhaps you can insert the phrase "a diet leading to obesity" instead.
Response #2:
We would like to thank the reviewers for their thoughtful comments and efforts toward improving our manuscript. We have changed it and inserted the phrase "a diet leading to obesity”. (Edit line 98)
#3 Current treatment - "but eventually, the degeneration.." can be changed to "but there is increased risk of further degeneration at operated or adjacent levels and the pain often recurs
Response#3:
Thank you so much for the reviewer’s comments. We have changed the sentence as the reviewer’s suggested. (Line 116 and 117)
#4 1 line 134 delete the extra "the"
Response#4:
We would like to thank the reviewers for their thoughtful comments and efforts toward improving our manuscript. We have deleted the extra word. (Edit line 135)
#5 Mesenchymal "several preclinical studies" should be followed by several references, not one reference [71].
Line 209 - did you mean "pre-senescent NPCs"?
Line 226 - there is no risk of graft rejection with autologous cells, so you need to specify this is true for "allogeneic" cells.
Response#5:
Thank you so much for the reviewer’s suggestions. We have added several references. (Edit line 204 and references 68,71-73)
Thank you so much for the reviewer’s comments. We made corrections according to the reviewer’s suggestions. (Edit line 210)
We are grateful for the reviewer’s comments. We have added some studies to specify it. (Line 230-239).
#6. 2 Mesenchymal - line 290-291 delete the phrase "of exosomes" and change "Similar in normal exosomes..." to "As with other exosomes, those derived from MSCs are also enhanced with..."
Response#6:
We are thankful for the reviewer’s suggestions. We have changed it and rewritten it according to the reviewer’s suggestions. (Edit lines 306 and 309)
#7. Application "vascularization of endothelial cells" - what does that mean?
Line 315 "human stem cells" - do you mean "human MSCs" ?
Line 330 - the sentence "Extensive research..." is a conclusion and not an introduction to the next section so just delete it.
Line 355 You start the sentence with "Abundant data" but end it with only two references! If the data were really abundant, then you need at least 3 or more references, not two.
Line 364, the sentence "Similarly, " should state if these are human BM-MSCs as in "Similarly, human BM-MSCs... " if that is true.
Line 367 - the sentence should end after MMP-3, followed by "These findings..."
Figure 2a - It should be made clear that this is a general diagram, not one related to a specific paper. Thus consider: "Figure 2. General diagram of experimental intradiscal injection of MSC-derived....."
Discussion: line 504 "Likewise, 'many' studies..." should be followed by 'many' references, not just Ref [139]!
Line 527 - "unsuitable environment for NPC proliferation" yet NPCs are naturally there! Isn't that like saying that a desert is an unsuitable environment for a cactus?
In general, the paper needs some editing, not necessarily from an English language expert, but from an expert in language and literature to reduce unnecessary words, phrases and sentences and make the paper more concise and better organized.
Response#7:
We are grateful for the reviewer’s comments. The term vascularization of endothelial cells indicates the process of growing blood vessels. This vascular system is necessary to provide adequate blood flow to cells, supplying oxygen and nutrients. In the study (Reference:113), the scientist suggests that exosomes from ADSCs that overexpress miR-21 potentially promote vascularization and show exosomes from their culture may be suitable for clinical effort in regenerative medicine.
Thank you so much for the reviewer’s comments. We mean to say human MSCs, and we have changed and written it. (Edit line 331)
We would like to thank the reviewers for their thoughtful comments and efforts toward improving our manuscript. We have deleted the sentence.
We are very grateful for the reviewer’s suggestions. We are thankful for pointing this out. We have added several references. (Edit line 371 and references 15,73, 82, 124-130)
Thank you so much for the reviewer’s comments. We have changed the sentence. (Edit line 379)
We are very thankful for the reviewer’s comments. We have changed the sentence and written it. (Edit line 382)
We are grateful for the reviewer’s suggestions. We have rewritten as the reviewer’s comments. (Edit line 493)
We are thankful for the reviewer’s suggestions. We have added several references. (Edit line 523 and references 147,149-155)
Thank you so much for the reviewer’s suggestions and great efforts toward improving our manuscript. We have followed the reviewer’s opinion and removed the sentence and changed it. (Edit line 548)
Thank you so much for the reviewer’s suggestions. We have reduced some unnecessary words and sentences.
Line 85,86: The flow of the sentence was unusual. We deleted the unnecessary word.
Line 149 -152: The flow of the sentence was unusual. We removed the unnecessary word and rewritten them.
We deleted the extra sentence from the discussion part and make it more organized. (Edit line 566, Reference:161)
Line 461-463: The flow of the sentence was unusual. We have deleted the extra word.
Line 553-555: The flow of the sentence was unusual. We have deleted the extra word to make concise.
Reviewer 2 Report
1- Compare and summarize the advantages and disadvantages of biological therapies in different studies.
2- Line 453, write the complete form of CCL545, CCL4, CCL3, CXCL1, and XCL1 first. Then use the abbreviation.
3- Combine the conclusion and future prospects in one section. Bring the conclusion first and then write the future prospects.
4- Some references are too old such as ref: 2, 23, and 24, it is suggested to replace them with some newly published papers.
5- The authors can use the following reference in this manuscript:
Sabbagh, F., Muhamad, I. I., Niazmand, R., Dikshit, P. K., & Kim, B. S. (2022). Recent progress in polymeric non-invasive insulin delivery. International Journal of Biological Macromolecules.
Author Response
Response to Reviewer 2 comments
#1 Compare and summarize the advantages and disadvantages of biological therapies in different studies
Response#1:
We are very grateful for the reviewer’s comments. We have made a table of the advantages and disadvantages of different biological therapies and their effects. (Edit lines 253-256)
#2 Line 453, write the complete form of CCL545, CCL4, CCL3, CXCL1, and XCL1 first. Then use the abbreviation
Response#2:
Thank you so much for the reviewer’s suggestions and great efforts toward improving our manuscript. We have completed the form and used the abbreviation. (Edit line 468-471 and 645-649)
#3 Combine the conclusion and future prospects in one section. Bring the conclusion first and then write the future prospects
Response#3:
Thank you so much for the reviewer’s comments. We have made corrections and written them. (Edit line 568 – 595)
#4 Some references are too old such as ref: 2, 23, and 24, it is suggested to replace them with some newly published papers.
Response#4:
We are grateful for the reviewer’s comments. We have replaced them with newly published papers. (Edit lines 34, 81, and 83)
#5 The authors can use the following reference in this manuscript:
Sabbagh, F., Muhamad, I. I., Niazmand, R., Dikshit, P. K., & Kim, B. S. (2022). Recent progress in polymeric non-invasive insulin delivery. International Journal of Biological Macromolecules.
Response#5:
Thank you so much for suggesting the reference. We have added the reference. (Edit line 530-532)
Round 2
Reviewer 1 Report
Abstract: Delete “In decades” and capitalize “mesenchymal”
Line 40: change “like” to “such as”
Line 43: Delete the sentence starting with “These modalities” and you can move reference (5) to the previous sentence if appropriate.
Line 86: Delete “are the main phenomena which”
Line 98: Change “a diet” to “excessive calories”
Line 117: Change “reoccurs” to “recurs.”
Line 152: Insert “a” in front of “suitable”
Line 230: Delete “by”
Line 232: Delete “the”
Line 235: Delete “by”
Line 236: Change “for” to “at” and “was” to “were”
Line 495: Change the first “the” to “a”
Line 549: Insert “a” in front of “complex”
Line 573: Change “indeterminate” to “unknown”
Line 580: Change “disc” to “discs”
Author Response
Response to Reviewer 1 Comments
Abstract: Delete “In decades” and capitalize “mesenchymal”
Line 40: change “like” to “such as”
Line 43: Delete the sentence starting with “These modalities” and you can move reference (5) to the previous sentence if appropriate.
Line 86: Delete “are the main phenomena which”
Line 98: Change “a diet” to “excessive calories”
Line 117: Change “reoccurs” to “recurs.”
Line 152: Insert “a” in front of “suitable”
Line 230: Delete “by”
Line 232: Delete “the”
Line 235: Delete “by”
Line 236: Change “for” to “at” and “was” to “were”
Line 495: Change the first “the” to “a”
Line 549: Insert “a” in front of “complex”
Line 573: Change “indeterminate” to “unknown”
Line 580: Change “disc” to “discs”
Response:
We are very grateful for the reviewer’s insightful comments on our paper. We have changed it as the reviewer’s suggested. (Edit line19)
Thank you so much for the reviewer’s suggestions. We have changed it as the reviewer’s suggested. (Edit line 40)
We would like to thank the reviewers for their thoughtful comments and efforts toward improving our manuscript. We have deleted the sentence and moved reference (5) to the previous sentence.
We are grateful for the reviewer’s comments. We have deleted as the reviewer’s suggested. (Edit line 85)
Thank you so much for the reviewer’s suggestions. We have changed the sentence as the reviewers suggested. (Edit line 97)
We are very thankful for the reviewer’s comments. We have changed the word. (Edit line 116)
We are very grateful for the reviewer’s suggestions. We are thankful for pointing this out. We have added the word as reviewers suggested. (Edit line 150)
We would like to thank the reviewers for their thoughtful comments and efforts toward improving our manuscript. We have deleted the word. (Edit line 229)
Thank you so much for the reviewer’s suggestions. We have deleted the word as the reviewer suggested. (Edit line 231)
We are thankful for the reviewer’s suggestions. We have deleted the word as the reviewer suggested. (Edit line 234)
We are very grateful for the reviewer’s suggestions. We have changed the word as the reviewers suggested. (Edit line 235)
Thank you so much for the reviewer’s suggestions. We have changed the word as the reviewers suggested. (Edit line 493)
Thank you so much for the reviewer’s suggestions and great efforts toward improving our manuscript. We have inserted a word. (Edit line 547)
We are thankful for the reviewer’s suggestions. We have changed the word as the reviewers suggested. (Edit line 571)
We would like to thank the reviewers for their thoughtful comments and efforts toward improving our manuscript. We have changed the word as the reviewers suggested. (Edit line 578)
